# The miR-26 family regulates early B cell development and transformation

Katharina Hutter[1], Silke E Lindner[1] , Constanze Kurschat[1], Thomas Rülicke[2] , Andreas Villunger[1,3,4] , Sebastian Herzog[1]

**MiRNAs are small noncoding RNAs that promote the sequence-specific repression of their respective target genes, thereby regulating diverse physiological as well as pathological processes. Here, we identify a novel role of the miR-26 family in early B cell development. We show that enhanced expression of miR-26 family members potently blocks the pre-B to immature B cell transition, promotes pre-B cell expansion and eventually enables growth factor independency. Mechanistically, this is at least partially mediated by direct repression of the tumor-suppressor *Pten*, which consequently enhances PI3K-AKT signaling. Conversely, limiting miR-26 activity in a more physiological loss-of-function approach counteracts proliferation and enhances pre-B cell differentiation in vitro as well as in vivo. We therefore postulate a rheostat-like role for the miR-26 family in progenitor B cells, with an increase in mature miR-26 levels signaling cell expansion, and facilitating pre-B to the immature B cell progression when reduced.**

## Introduction

Early B cell development in the bone marrow is a strictly regulated process characterized by sequential rearrangements of gene segments within the B cell receptor (BCR) heavy and light chain genes. Initiated first on the Ig heavy chain locus at the pre-pro-B–cell stage [1], D-to-J followed by productive V-to-DJ gene segment recombination gives rise to a mature Ig mu heavy chain that is expressed on the cell surface in form of the pre-BCR. The pre-BCR, which defines the pre-B cell stage, is composed of two mu heavy chains that pair with the surrogate light chain components VpreB and lambda5 [2]. Ligand-independent autonomous crosslinking of the pre-BCR serves as a crucial checkpoint in development and orchestrates the clonal expansion of pre-B cells [3]. This is followed by a transition into resting small pre-B cells, which undergo recombination of variable and joining gene segments within the Ig

light chain (LC) genes and eventually express the mature BCR on the cell surface [4].

This stepwise progression, with alternating phases of proliferation (in early pro-B and pre-B cells) and differentiation involving DNA rearrangements (in late pro-B and small pre-B cells), has to be tightly regulated to preserve genomic stability, as errors within these developmental processes can promote pre-B cell leukemia, immunodeficiency and autoimmune diseases [5, 6, 7, 8, 9]. This is mainly ensured by the interplay of the IL-7 receptor (IL-7R) and the pre-BCR, both of which dominate at a certain time throughout development [10]. In pro-B cells, signaling through the IL-7R and the JAK/STAT as well as the PI3K-AKT modules is crucial for proliferation and survival [11]. Moreover, these pathways actively suppress premature recombination of gene segments at the LC loci, both on the level of RAG enzymes as well as by orchestrating the accessibility of the LC locus [11, 12, 13]. In synergy with signals emanating from the IL-7R, expression of the pre-BCR drives an initial proliferative phase with four to five cell divisions, and then promotes cell cycle exit by the activation of the SYK-SLP-65 (Src homology domain-containing leukocyte protein of 65 kD, also referred to as BLNK) module. Signaling via SLP-65 terminates or at least dampens PI3K-AKT signaling, which in turn activates the transcription factors FOXO1 and PAX5 and their downstream mediators to initiate LC gene recombination [14]. This is supported by concomitant induction of the RAS-ERK MAPK module, which allows full *Rag1/2* gene expression and also enables LC locus accessibility [15]. The switch from IL-7R to pre-BCR dominance is further enhanced by CXCR4 signaling, which repositions B cell precursors from IL-7[high] to IL-7[low] niches within the bone marrow microenvironment. Moreover, recent data indicate that CXCR4 signaling also plays a direct role in cell cycle exit, B cell survival and regulation of LC gene recombination [16].

However, even with this complex network of three receptors and their downstream signaling cascades in place, it appears fair to say that the precise regulation of early B cell development is only partially solved. Indeed, the discovery of noncoding RNAs, in particular miRNAs, has added a layer of posttranscriptional regulation to these processes that is far from completely understood. MiRNAs are small non-coding RNAs of about 21–24 nucleotides that

[1]Institute of Developmental Immunology, Biocenter, Medical University Innsbruck, Innsbruck, Austria   [2]Department of Biomedical Sciences and Ludwig Boltzmann Institute for Hematology and Oncology, University of Veterinary Medicine Vienna, Vienna, Austria   [3]CeMM Research Center for Molecular Medicine of the Austrian Academy of Sciences, Vienna, Austria   [4]Ludwig Boltzmann Institute for Rare and Undiagnosed Diseases, Vienna, Austria

Correspondence: Sebastian.herzog@i-med.ac.at

promote the sequence-specific repression of their respective target genes (17). It has been estimated that more than two thirds of all coding genes, and consequently every biological process, are under control of miRNAs (18). This is also true for B cell development, which is significantly impaired from an early stage on in the absence of the miRNA processing machinery (19, 20, 21). More specifically, several miRNAs have already been described to fine–tune transition through the different B cell developmental stages, among them, for example, the miR-17-92 cluster, which is required for proper pro-/pre-B cell survival through repression of pro-apoptotic *Bim* and most likely also by ensuring proper PI3K signaling (22). Likewise, in vitro data suggest that the miR-15 family, best known for its tumor-suppressing function in chronic lymphocytic leukemia, regulates the interplay of proliferation and differentiation in early B cells (23). Later stages of the B cell life cycle are, for example, regulated by the miR-29 family and by miR-155, which control maintenance and terminal differentiation of mature B cells, respectively (24, 25, 26). Overall, however, a definitive function in B cell development has only been defined for relatively few miRNAs, suggesting that additional regulators will emerge upon thorough analyses.

In this study, we have followed up on an in vitro screen that intended to identify miRNAs involved in malignant transformation in B cell progenitors (27). Our findings reveal that the pre-B to immature B cell transition is potently blocked by enhanced expression of miR-26 family members miR-26a and b, both of which are found at relatively high levels in various immune cell subsets (28). Notably, aberrant expression of miR-26 not only inhibits B cell development, but promotes pre-B cell expansion and growth factor independency, pointing towards a proto-oncogenic function. Mechanistically, we identify the tumor-suppressor *Pten* as a direct target of miR-26 in this context and show that its repression enhances PI3K-AKT signaling. Conversely, limiting miR-26 activity in a more physiological loss-of-function approach counteracts proliferation and enhanced pre-B cell differentiation in vitro as well as in a mouse model. Together, our data reveal a novel role of the miR-26 family in regulating the pre-to immature B cell transition.

## Results

### Overexpression of miR-26 family members suppresses pre-B cell differentiation

To identify miRNAs that could potentially modulate early B cell development, we have recently performed a miRNA overexpression screen where we exploited the established B cell precursor cell line 1676 which is deficient for the adaptor proteins SLP-65 and LAT (linker for activation of T cells) (27). Because of their genetic constitution, 1676 cells are arrested at the pre-B cell stage, constitutively express and depend on the pre-BCR and proliferate in the presence of their growth factor IL-7. However, withdrawal of IL-7 triggers pre-BCR down-regulation and initiates recombination of gene segments at the κ LC gene, which gives rise to BCR⁺ immature B cells and thus recapitulates key processes of early B cell development (Fig 1A).

Within this setup, expression of a retroviral library comprising about 60 pri-miRNAs followed by IL-7 withdrawal and assessment

of the percentage of κ⁺ cells led to the identification of several miRNAs either promoting or suppressing pre-B to immature B cell differentiation (Fig 1B). The strongest enhancement of differentiation was observed upon expression of miR-15 family members, which is in correspondence with our previous work (23). The most severe block of differentiation, on the other hand, was induced by overexpression of the two miR-26 family members miR-26a and miR-26b (Fig 1B, blue columns). As miR-26a and miR-26b share the same seed sequence and thus are expected to regulate the same set of target genes, from here on the construct encoding for miR-26a was used for all following miR-26 overexpression experiments. To validate the suppressive function of miR-26 in pre-B cell differentiation, we exogenously expressed miR-26a also in bone marrow-derived primary B cell progenitors as well as in another pre-B cell line arrested at the pre-B cell stage (wk3). In analogy to the screen, miR-26a overexpression resulted in a significant reduction of differentiation into BCR⁺ immature B cells upon IL-7 withdrawal in both settings, albeit less pronounced in the primary culture (Fig 1C). Nevertheless, these data clearly indicate that aberrant miR-26 levels interfere with the pre-B to immature B cell transition. This was further supported by quantitative RT-PCR analysis of 1676 control and miR-26a-expressing cells cultured either with or without IL-7 for 48 h. Here, key transcripts normally induced by IL-7 withdrawal, such as *Rag1*, *Rag2*, *Ikzf1* (Ikaros), and *Ikzf3* (Aiolos), were not or less strongly up-regulated when miR-26a was overexpressed (Fig S1A and B).

One prerequisite for pre-B cell differentiation is the shutdown of the cells' proliferative program to enable gene segment recombination at the LC loci (11). We therefore wondered whether the reduced differentiation mediated by miR-26 might be the consequence of an enhanced proliferative capacity in pre-B cells. Indeed, both in bone marrow-derived primary B cell progenitors as well as in the pre-B cell line, high miR-26 levels conferred an advantage in a competitive setting under IL-7⁺ steady-state conditions (Fig 1D). Because this competition assay cannot distinguish between cell survival and proliferation effects, we directly investigated these processes through EdU labeling, DNA content staining and by measuring chromatin condensation (Fig S1C–E). Surprisingly, this did neither reveal a clear proliferative advantage of miR-26a–overexpressing cells under steady-state conditions, nor did it demonstrate a significant survival benefit, suggesting that these approaches may not be sensitive enough to detect small differences. Thus, it is unclear whether the enrichment of miR-26-expressing cells is an effect of enhanced proliferation, reduced apoptosis or both. However, when assessing cell death upon depletion of IL-7, which is a trigger of apoptosis in this in vitro system, miR-26a expression had a clear pro-survival effect (Fig 1E). To clarify whether this is restricted to growth factor withdrawal or rather a general anti-apoptotic feature, we further tested the cellular response to etoposide, a potent inducer of DNA strand breaks. Using the percentage of miR-26a expressing cells over time as a readout, we observed an enrichment and thus selective advantage of treated cells compared with the untreated controls, indicating an anti-apoptotic effect of miR-26a also in this setting (Fig 1F). Together, these data show that the miR-26 family not only opposes pre-B cell differentiation, but also supports early B cell expansion and survival under growth factor limiting conditions or toxic stimuli.

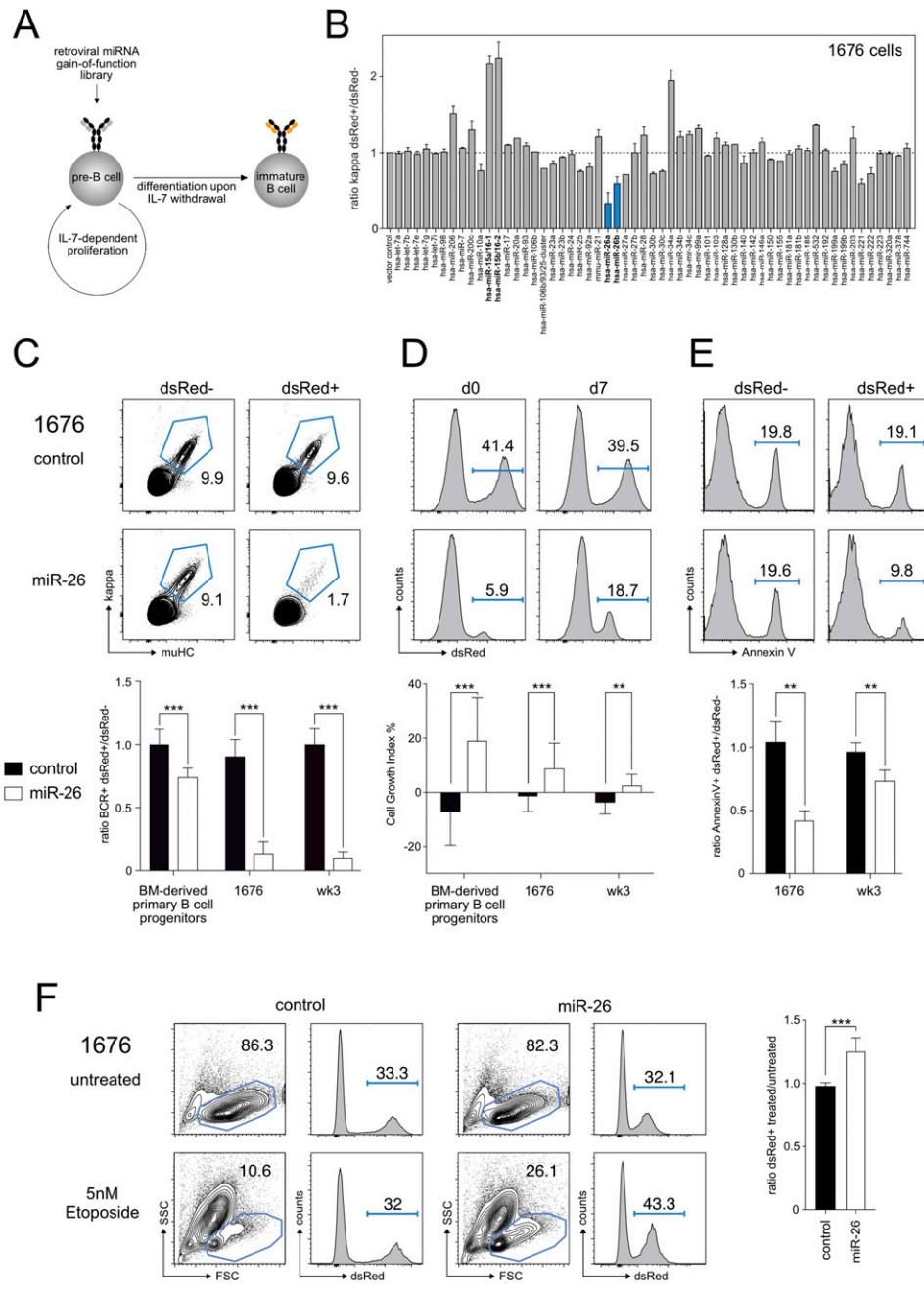

**Figure 1. Overexpression of miR-26 family members drives pre-B cell expansion, blocks differentiation, and protects from apoptosis.**

**(A)** Schematic illustration of the miRNA overexpression screen to investigate pre-B cell differentiation upon IL-7 withdrawal. **(B)** 1676 pre-B cells transduced with miRNA overexpression constructs as indicated were cultured in the absence of IL-7 for 48 h to induce differentiation into immature B cells. Differentiation was analyzed by measuring surface expression of the kappa light chain as part of the mature B cell receptor. The bar graph depicts the ratio of dsRed+ (transduced) and dsRed– (non-transduced) kappa⁺ immature B cells within one sample (N = 4). **(C)** 1676 pre-B cells, wk3 pre-B cells, or primary B cell progenitors transduced for overexpression of miR-26a or an empty control vector were cultured without IL-7 for 48 h to induce differentiation. Representative FACS plots of an experiment with 1676 pre-B cells compare the percentage of kappa⁺muHC⁺ mature B cells in transduced (dsRed+) and non-transduced (dsRed–) cells within the same sample. The bar graph provides the statistical analysis of at least four independent experiments using primary B cell progenitors, the 1676 or the wk3 pre-B cell line, respectively. **(D)** 1676 pre-B cells transduced for overexpression of miR-26a or an empty control vector with dsRed as a fluorescent marker were cultured with IL-7 for 7 d. Representative FACS plots show the change in the percentage of dsRed+ cells over time. The bar graph shows the statistical analyses of the corresponding experiments in primary progenitors, in 1676 and in wk3 cells. Cell growth indices were calculated based on at least four independent experiments. **(E)** MiR-26a overexpression protects against apoptosis upon growth factor withdrawal. 1676 pre-B cells were transduced with the indicated constructs and cultured in the absence of IL-7 for 48 h. Numbers in histograms indicate the percentage of apoptotic cells defined as PI/ DAPI–negative and Annexin V–positive. The bar graph displays the analyses of the corresponding experiments in both 1676 and in wk3 cells. **(F)** 1676 pre-B cells transduced with indicated constructs were treated with 5 nM etoposide for 20 h. Histograms depict the percentage of transduced (dsRed+) cells before and after treatment. **(C, D, E, F)** Bar graphs show mean + SD and represent at least four (C, D), eight (E) or seven (F) independent experiments; groups were compared by paired *t* tests; **P < 0.01, ***P < 0.001.

Notably, such features are typical for proto-oncogenes, which made us wonder about the consequences of miR-26 expression on the cellular level.

### Aberrantly high levels of miR-26a can transform pre-B cells

When expressing oncogenes such as BBL or constitutively active RAS, *SLP-65⁻/⁻LAT⁻/⁻* pre-B cell lines display ongoing proliferation and survival even in the absence of their growth factor IL-7 (27). This mimics a transformation-like state and recapitulates the key phenotype of pre-B leukemia, making this system a suitable tool to assess a gene's oncogenic potential. Exploiting this feature, monitoring pre-B cells under conditions lacking IL-7 revealed that overexpressing miR-26a, but not its respective seed mutant (miR-26aᵐᵘᵗ), was indeed sufficient to drive continuous expansion and rendered cells growth factor-independent (Fig 2A). Notably, when repeated in bone marrow-derived primary B cell precursors, aberrant expression of miR-26a on its own was not able to transform cells (data not shown), most likely reflecting that additional hits, such as the loss of pre-BCR signaling components in the tested pre-

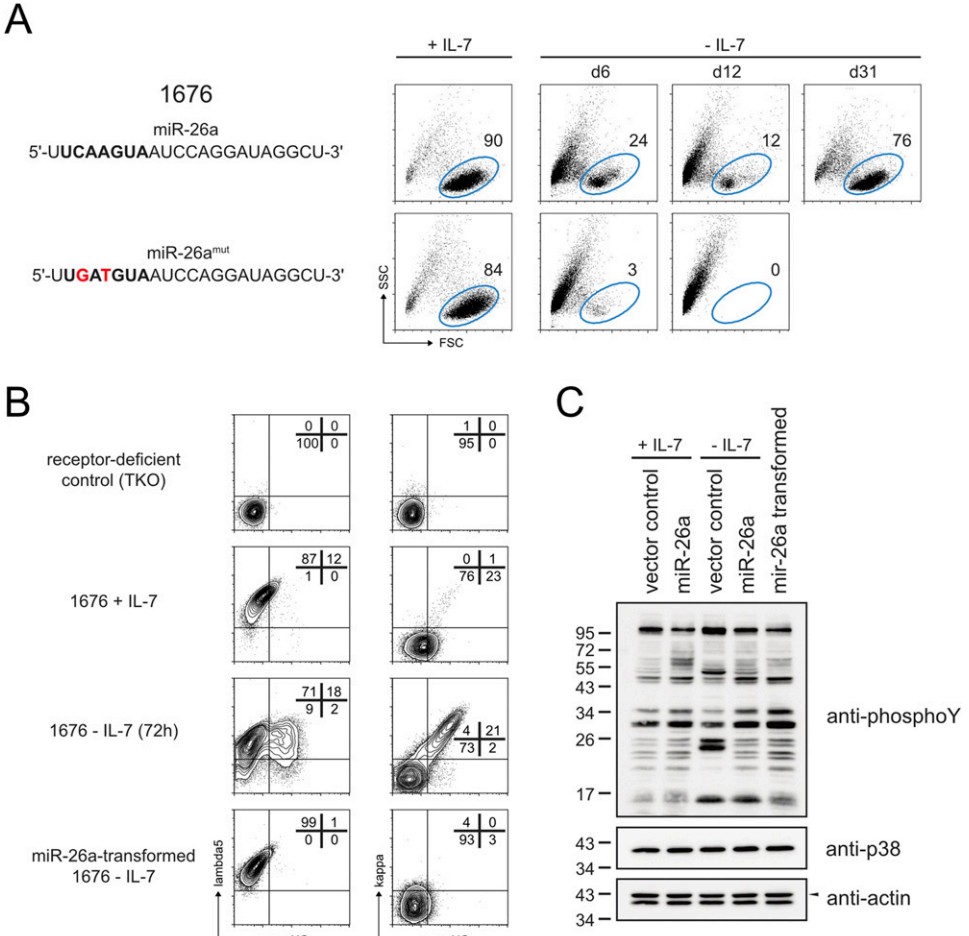

**Figure 2.  Aberrantly high miR-26a levels promote growth factor independence.**
**(A)** 1676 pre-B cells overexpressing miR-26a or a variant in which two nucleotides within its seed region were mutated (bold letters, point mutations in red) were cultured in the absence of IL-7 for an extended time. Representative FACS plots show the percentage of living pre-B cells as defined by FSC/SSC parameters at different time points. Representative of three independent experiments. **(B)** Parental 1676 pre-B cells cultured with IL-7 or without IL-7 and an established IL-7–independent clone of miR-26a-expressing cells were analyzed for pre-B cell receptor (BCR) (lambda5) and mature BCR (kappa) expression. A differentiation-defective pre-B cell line lacking a pre-BCR (TKO) was used as a negative control. Numbers represent the percentages of cells within the respective gates. Shown plots are representative of three independent experiments. **(C)** Anti-phosphotyrosine Western blot comparing 1676 pre-B cells cultured with IL-7 or after IL-7 withdrawal for 12 h and expressing the constructs as indicated with transformed, IL-7–independent miR-26⁺ cells. Actin (upper band, marked by a black arrow) and p38 serve as loading controls.

B cell lines, are required to fully establish a leukemic phenotype. In correspondence, even in the cell lines, most of the miR-26a–expressing cells underwent apoptosis upon IL-7 withdrawal (Fig 2A), and only a fraction eventually became independent of the growth factor. This points to a strong selection within the heterogeneous population, and likely illustrates the dependence on additional oncogenic mutations. To investigate how miR-26a transforms pre-B cells, we took a closer look at pre-BCR expression, since continuous signaling from the pre-BCR has been shown to promote proliferation and leukemic progression (29). Upon IL-7 withdrawal, control cells down-regulated their pre-BCR as measured by lambda5, which was accompanied by gene segment recombination at the κ LC loci and expression of the mature BCR on the cell surface (Fig 2B; third line). In contrast, miR-26a-expressing cells retained their pre-BCR⁺ phenotype (Fig 2B; fourth line). Interestingly, an assessment of their intracellular phospho-tyrosine signaling status revealed that miR-26 expressing cells cultured without IL-7 for 12 h as well as transformed cells were indistinguishable from their control counterparts cultured in the presence of IL-7 (Fig 2C), suggesting that these cells can maintain a proliferative signaling pattern. This raised the question whether continuous pre-BCR expression and its downstream signaling is causative for or simply correlates with the leukemic phenotype. To investigate this, we tested two shRNAs for silencing the critical pre-BCR subunit CD79b, both of which

significantly reduced pre-BCR surface expression compared with the control (Fig 3A). Exploiting the more potent shCD79b #1, transformed cells lacking CD79b had a clear competitive disadvantage and became significantly reduced over time (Fig 3B), indicating the positive impact of pre-BCR signaling. This pre-BCR dependence was further confirmed by pharmacological inhibition of SYK, one of the key kinases critical for initiation of pre-BCR and BCR signaling (30). Here, transformed cells lacking pre-BCR signals displayed a strongly reduced cell size, which is an indicator for cell cycle arrest, both in the presence as well as in the absence of their growth factor IL-7 (Fig 3C), and eventually underwent apoptotic cells death. Hence, we conclude that continuous pre-BCR signaling is required to keep miR-26a–expressing cells in their leukemic state.

Furthermore, we investigated whether transformation requires continuous miR-26a expression, also referred to as oncogene addiction, or whether miR-26a is only required to initiate the leukemic state. In detail, we inhibited miR-26a activity in established, IL-7–independent clones using a miRNA sponge, an mRNA containing multiple concatemeric repeats of miR-26–binding sites in the 3′-UTR (Fig 3D; reference 31). When transcribed, the miR-26 sponge is supposed to sequester the RISC-incorporated miR-26 family members and thereby induces a derepression of endogenous target genes. Indeed, expression of a miR-26–specific sponge, but not of a control, resulted in a clear derepression of a reporter

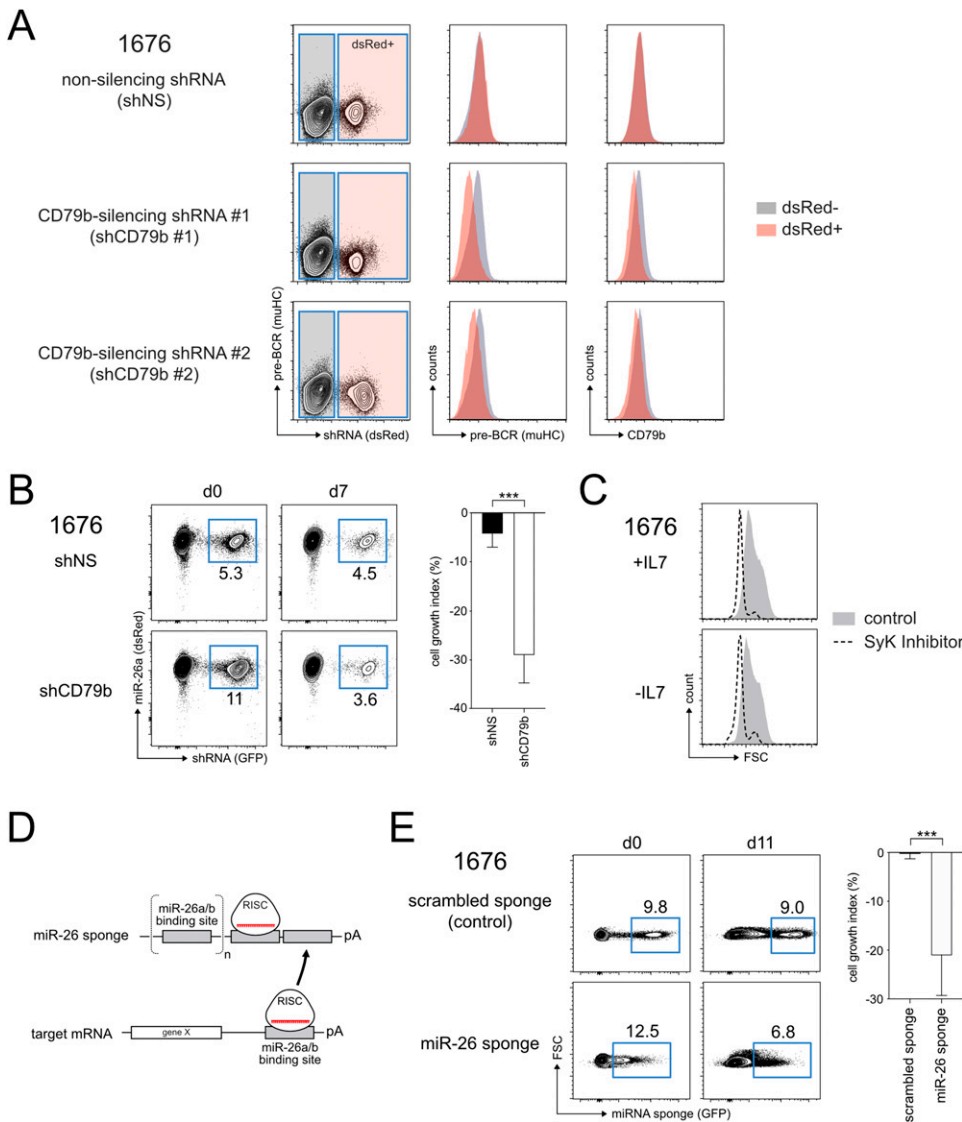

**Figure 3. Transformed pre-B cells are addicted to pre-BCR signaling and depend on high miR-26 expression.**
**(A)** Flow cytometric analysis indicating a clear reduction in surface pre-BCR expression upon knockdown of CD79b in 1676 cells. **(B)** Pre-BCR signaling is indispensable for miR-26a–transformed pre-B cells. MiR-26⁺ IL-7–independent 1676 pre-B cells were stably transduced with a control non-silencing shRNA or with an shRNA targeting CD79b. Representative FACS plots show the percentage of transduced (GFP+) cells at indicated time points. Bar graphs summarize mean + SD of four independent experiments. **(C)** Transformed 1676 cells cultured with or without IL-7 were treated with DMSO only (grey histogram) or with the Syk inhibitor R406 (dashed line) for 48 h. The FSC pattern serves as an indirect marker for proliferation. **(D)** Schematic illustration of the mechanisms underlying miRNA-sponge mediated sequestration of specific RISC complexes. **(E)** IL-7–independent pre-B cells are addicted to high miR-26 levels. Transformed 1676 cells cultured in the absence of IL-7 were transduced with a control (scrambled) or a miR-26 sponge construct. The percentages of transduced (GFP+) cells were measured at d0 and d7 as indicated by representative contour plots. Bar graphs summarize the mean growth indices + SD of five independent experiments. Groups were compared by a paired $t$ test; ***$P < 0.001$.

sensing the activity of endogenous miR-26a and b (Fig S2). In contrast to the scrambled sponge control, expression of the miR-26 sponge in transformed cells conferred a competitive disadvantage, indicating that cells in their transformed, proliferative state are selected for high levels of miR-26 (Fig 3E).

Together, these data demonstrate that aberrant miR-26a expression arrests cells at a pre-BCR⁺ stage and enables growth factor independence, and that the maintenance of this transformation-like state critically depends on the miRNA and on continuous pre-BCR downstream signaling. This raised the question how miR-26a mediates this profound reprogramming on the molecular level.

### Mir-26 overexpression activates PI3K signaling by repression of *Pten*

MiRNAs mediate their function by regulating networks of target genes, often resulting in complex phenotypes as also observed

here. Nevertheless, we anticipated that aberrant miR-26 regulation might impact one of the key signaling pathways controlling the pre-B checkpoint, thereby interfering with proper pre-B to immature B cell transition. To investigate this in detail, we performed transcriptome analysis comparing miR-26a overexpressing cells with control pre-B cells. Genes that showed at least 20% repression upon miR-26a overexpression were filtered for conserved miR-26 binding sites using different prediction tools, which resulted in a total of 187 putative target genes (Fig 4A and Table S1). To validate some of these putative hits as direct targets of miR-26a, we expressed the 3′-UTRs of selected genes together with GFP and assessed whether miR-26a, but not its seed mutant, affects reporter expression (Fig 4B). Indeed, several 3′-UTR constructs were directly repressed by miR-26, among them *Pten* (Phosphatase and tensin homolog), a known tumor suppressor and negative regulator of the PI3K-AKT signaling axis (Fig 4B, blue columns). This raised our interest because PI3K signaling has been demonstrated to instruct

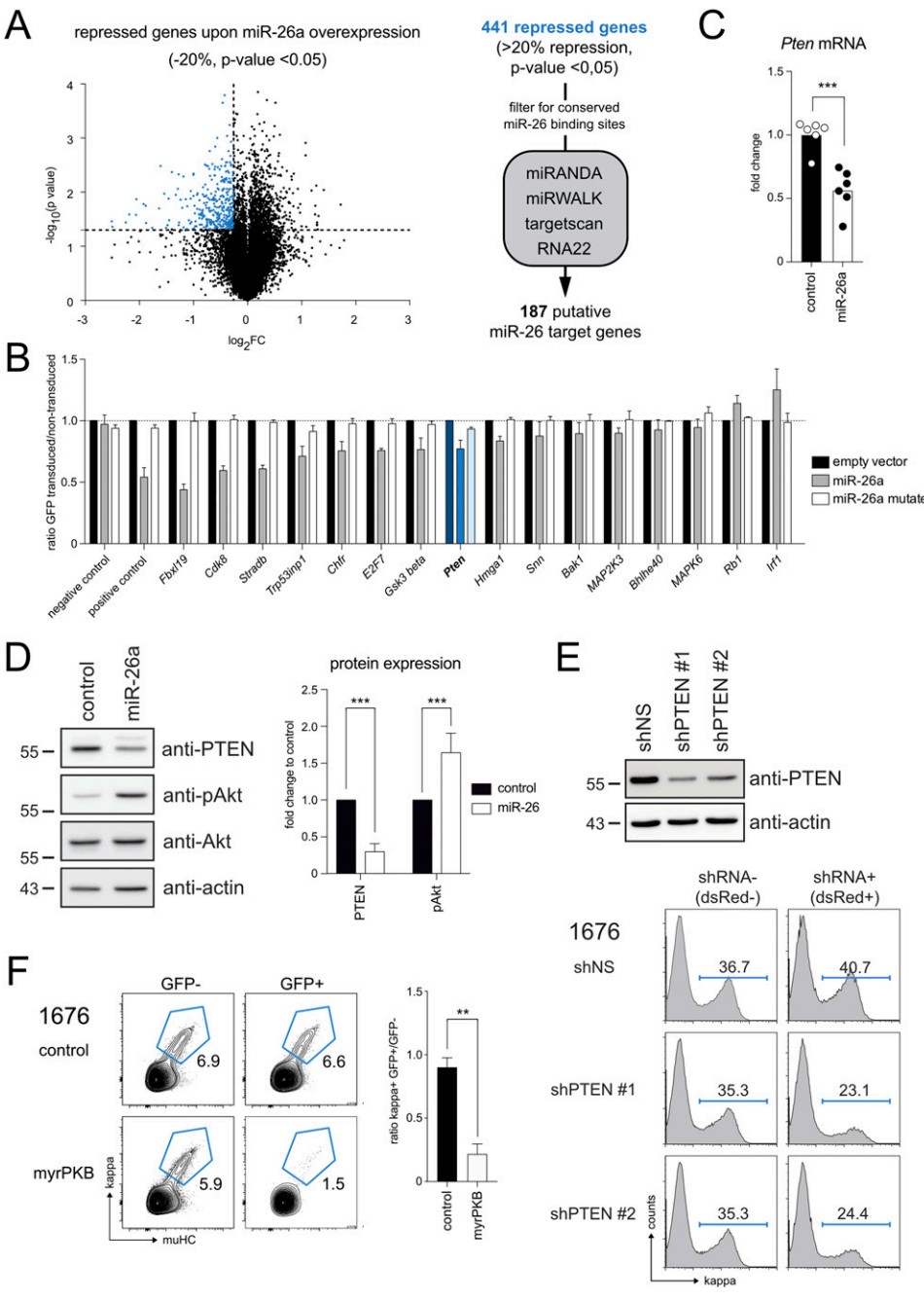

**Figure 4. Mir-26 overexpression activates PI3K signaling through repression of *Pten*.**
**(A)** Volcano plot showing the transcriptome changes in miR-26a–expressing 1676 pre-B cells compared with the respective control (two independent experiments). The horizontal dashed line marks the *P*-value cutoff, the vertical line depicts a repression by 20% or more. Genes that are significantly down-regulated upon miR-26a expression are displayed in blue. The flow chart on the right summarizes the strategy for miR-26a target identification. **(B)** Putative target genes identified by microarray analysis were validated by 3′UTR reporter assays. 1676 pre-B cells expressing GFP reporters containing the 3′-UTRs of putative target genes were transduced with an empty vector, a miR-26a construct or a seed mutant of miR-26a (see Fig 2A). Bar graphs show the GFP mean fluorescence intensity comparing transduced and non-transduced cells as analyzed by flow cytometry and normalized to the empty vector. A reporter without a 3′-UTR and a reporter with a perfect complementary miR-26a site served as negative and positive controls, respectively. Depicted results are representative for at least four independent experiments. **(C)** Quantitative RT-PCR of *Pten* levels in 1676 pre-B cells overexpressing miR-26a or a control construct. The bar graph depicts *Pten* levels normalized to the control construct. **(D)** Western blot analysis determining protein levels of PTEN, phospho-AKT (pAkt), AKT, and beta-actin as a loading control. The bar graph summarizes PTEN and pAKT levels normalized to the empty vector control. Bar graphs (C, D) summarize mean + SD of six independent experiments. **(E)** Knockdown of *Pten* mimics the suppression of pre-B cell differentiation induced by miR-26a. Western blot analysis to confirm shRNA-induced knockdown of *Pten*. Histograms depict the percentages of immature (kappa⁺) pre-B cells upon transduction with shRNAs against *Pten* and IL-7 withdrawal for 48 h. Representative of two independent experiments. **(F)** Enhanced AKT signaling compromises pre-B cell differentiation. 1676 pre-B cells were transduced with a myristoylated form of AKT and cultured in the absence of IL-7 for 48 h. Differentiation of transduced (GFP+) and non-transduced (GFP−) cells was compared based on kappa expression. The bar graph represents mean + SD of four independent experiments. For statistical analysis, groups were compared by an unpaired *t* test; \*\**P* < 0.01, \*\*\**P* < 0.001.

precursor B cell expansion and to orchestrate gene expression at the pre-B cell stage (32, 33), making it a critical regulator in early B cell development. Inhibition of PI3K, on the other hand, has been shown to be sufficient to induce differentiation of pre-B cells even in the presence of IL-7 (14). It is therefore tempting to speculate that PI3K pathway activation needs to be turned down or shut off to enable the pre-B to immature B cell transition, and vice versa, that failure to lower PI3K signaling below a certain threshold may counteract this differentiation process.

Validating the microarray data, qRT-PCR analysis of an independent dataset confirmed that aberrant miR-26a expression reduces the level of *Pten* transcripts (Fig 4C). This was further supported by Western blot analysis, which showed a decrease in PTEN protein levels (Fig 4D). Notably, the reduction in PTEN was accompanied by an increase in phospho-AKT^S473 levels, which can serve as an indicator for PI3K pathway activation (Fig 4D). Thus, miR-26–mediated repression of *Pten* has clear functional consequences, that is, it activates the PI3K-AKT cascade, which may at least partially explain the miR-26–mediated pre-B cell phenotype described above. To test this in more detail, we used two alternative methods to activate the PI3K-AKT pathway independent of aberrant miR-26a expression,

anticipating that this would interfere with pre-B to immature B cell differentiation.

First, we performed shRNA-mediated *Pten* knockdown and could indeed observe reduced pre-B cell differentiation compared with controls upon IL-7 withdrawal (Fig 4E). To test whether this effect is in fact driven by AKT activation, and not by a PI3K-independent function of PTEN, we furthermore expressed a myristoylated variant of AKT in pre-B cells. This modification recruits AKT to the plasma membrane independent of upstream signaling events and in consequence renders the kinase constitutively active, thereby mimicking a hyper-activation of the PI3K pathway. Here, expression of myristoylated AKT recapitulated the shPTEN phenotype, that is, a reduced percentage of differentiated immature B cells upon IL-7 withdrawal (Fig 4F), albeit more pronounced because of the massive activation of AKT.

Together, this identifies PI3K signaling as a central pathway deregulated by miR-26a overexpression and suggests that *Pten* is a key target involved in the miR-26-induced block of early B cell development. Noteworthy, however, neither knockdown of *Pten* nor expression of myristoylated AKT were able to transform pre-B cell lines on their own (data not shown), indicating that the deregulated network underlying the miR-26–mediated phenotype is more complex and most likely supported by repression of additional target genes.

### Knockdown of endogenous miR-26 family members enhances pre-B cell differentiation

Having identified a block in early B cell development upon aberrant overexpression of miR-26a, we wanted to know whether a knockdown of the endogenous miR-26 family would eventually result in the reciprocal phenotype. If so, this would indicate a role of the miR-26 family not only in a potential oncogenic setting, but also under physiological conditions. Indeed, when interfering with endogenous miR-26a and b expression by their sequestration with a miR-26 family-specific sponge construct, we found an increase in pre-B cell differentiation upon withdrawal of IL-7 (Fig 5A). Notably, this phenotype was not restricted to different pre-B cell lines, but was also robustly observed in bone marrow-derived primary B cell progenitors. Furthermore, pre-B cells with reduced levels of functional miR-26 had a competitive disadvantage and got lost over time in both cellular systems (Fig 5B). Hence, the activity of the miR-26 family in early B cells determines cellular behavior in terms of cell expansion and differentiation: On one end of the spectrum, high miR-26 levels promote cell proliferation and/or survival and block the pre-B to immature B cell transition, whereas on the other end a reduction of the physiological miR-26 levels limits expansion and enhances pre-B cell differentiation in vitro. This made us wonder whether the same phenotype would also manifest in a more complex in vivo environment. Given that the miR-26 family consists of three different members miR-26a-1, miR-26a-2, and miR-26b (encoded on mouse chromosomes 9, 10, and 1, respectively), rendering traditional targeting tedious, we recapitulated our in vitro approach and generated a conditional miR-26 loss-of-function sponge model as well as a corresponding scrambled sponge control, both expressing GFP as a fluorescent marker. To test the functionality of the miR-26

sponge, we exploited a mechanism referred to as target-directed miRNA degradation (TDMD; reference 34). TDMD describes the phenomenon that a near-perfect complementary binding of a miRNA to its target sequence not only reduces the level of the target mRNA, but also of the miRNA itself. A sponge construct therefore not only functionally blocks miRNA activity, but can also reduce the level of the targeted miRNAs. Indeed, a qRT-PCR analysis of GFP-positive B cells sorted from the spleen of scrambled sponge and miR-26 sponge animals demonstrated a clear reduction of miR-26a and b in the latter (Fig 6A). Supporting sponge function, we furthermore found that miR-26 sponge-positive cells displayed significantly lower GFP levels compared with the control (Fig 6B), likely reflecting partial repression of the sponge because of efficient sequestration of miR-26a/b-loaded RISC complexes.

Having observed a clear effect of the sponge on mature miRNA levels, we wondered about its impact on B cell development. Interestingly, the pro- and pre-B cell compartment (AA4.1$^+$ IgM$^-$) in the bone marrow was strongly reduced at expense of the immature population (AA4.1$^+$ IgM$^+$; Fig 6C and D). Given that our in vitro data indicated an enhanced progression of pre-B cells towards the immature stage upon miR-26 sequestration, we interpret these in vivo findings in a similar way: pro- and pre-B cells appear to differentiate faster or more efficient, possibly also combined with a less extensive expansion at the pre-BCR checkpoint, resulting in a relative increase in immature B cells. Notably, later stages in the splenic compartment did not show any perturbations upon sequestration of miR-26 family members (Fig S3), making the alternative scenario in which the accumulation of immature B cells is due to a block in further maturation unlikely.

### Endogenous miR-26 likely functions through multiple, weakly repressed target genes

Having identified *Pten* as an important miR-26 target in the overexpression setting, with a likely role in establishing the block in pre-B cell differentiation, we wondered whether the reciprocal miR-26 loss-of-function phenotype might be driven by increased *Pten* expression and consequently reduced PI3K signaling. However, *Pten* levels were altered neither in cell lines expressing the miR-26 sponge nor in pre-B cells sorted from sponge mice, and correspondingly, we did not observe a clear reduction in phosphorylated AKT (Figs 7A and S4). We therefore hypothesized that other target genes must be responsible for the observed phenotype, and investigated this in more detail. In particular, we adopted the same strategy as for miR-26 overexpression and performed a transcriptome analysis of miR-26 sponge versus control cells, expecting to identify genes that become derepressed by the former. Strikingly and in contrast to the overexpression approach, sequestration of miR-26 family members by the sponge had only minimal effects on the transcriptome (Fig 7B), with basically very few significantly up- or down-regulated genes. Anticipating a rather modest effect of miR-26a and b on individual target genes, we therefore adapted our initial filtering criteria and selected for genes with an up-regulation by at least 20% at a *P*-value of 0.1 or lower. This gave rise to a total of 206 genes, and a filter for conserved miR-26 binding sites reduced

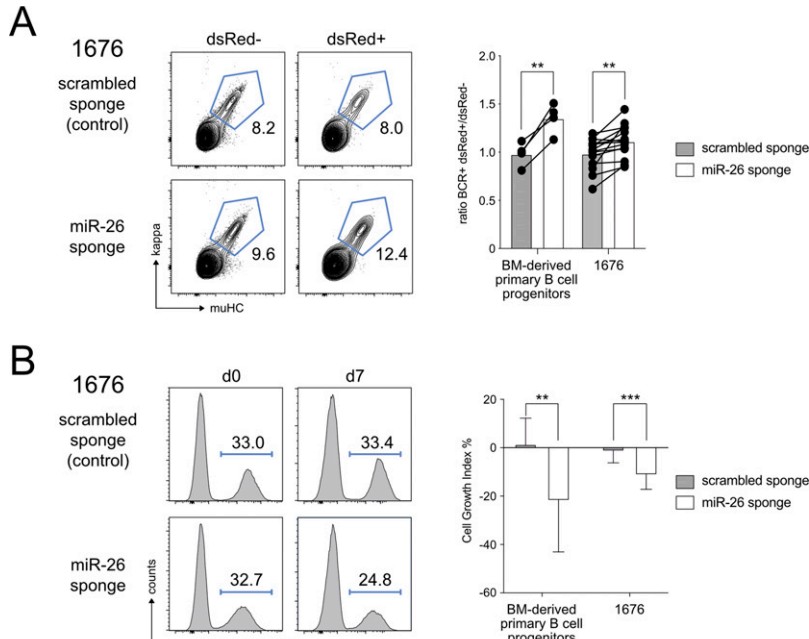

**Figure 5. Knockdown of endogenous miR-26a and b counteracts pre-B cell expansion and enhances pre-B to immature B cell differentiation in vitro.**

**(A)** 1676 pre-B cells transduced either with a scrambled sponge (control) or with a miR-26 sponge construct were cultured without IL-7 for 72 h to induce differentiation. Contour plots depict the percentage of immature B cells as defined by muHC⁺kappa⁺ in transduced (dsRed+) and non-transduced (dsRed−) cells within one sample. The bar graph summarizes the ratio of immature B cells among transduced and non-transduced cells in primary bone marrow-derived B cell precursors and in the 1676 cell line. Lines connect the data points of the independent experiments. **(B)** 1676 pre-B cells expressing the miR-26 sponge or a scrambled sponge construct were cultured with IL-7 for 7 d. Representative histograms show the change in the percentage of transduced (dsRed+) cells over time. The bar graph provides a statistical analysis based on the cell growth index for the pre-B cell line 1676 and for primary B cell progenitors. Graphs show mean + SD and represent at least four independent experiments (A, B); groups were compared by a paired t test; **P < 0.01, ***P < 0.001.

this list to 65 putative miR-26 targets (Fig 7B and Table S2). Surprisingly, the overlap between the identified targets by gain-of-function, that is, miR-26 overexpression, and the sponge-mediated loss-of-function was extremely small, with only five genes identified by both approaches (Table S3). Although possibly relevant to general miR-26 biology, none of these genes has been implicated in lymphocyte biology or B cells in particular, and most likely cannot explain the B cell phenotype. It therefore remains unclear by which target genes miR-26 enhances pre-B cell expansion and restricts the pre-B to immature B cell differentiation, respectively.

## Discussion

In this study, we have exploited an unbiased miRNA overexpression screen to identify miRNAs with a role in early B cell development. This led to the identification of the miR-26 family, comprising miR-26a and miR-26b, which we found to interfere with the pre-B to immature B cell transition in vitro.

Physiological functions that have been described for this miRNA family include the regulation of muscle development, glucose and lipid metabolism and adipogenesis (35, 36, 37), but surprisingly little is known about the role of miR-26a and b in immune cell development and function despite their relatively high expression within these tissues (28, 38). Aberrant expression of miR-26 family members, however, has been reported in several hematopoietic and non-hematopoietic malignancies. Interestingly, the prevailing picture does not define the miR-26 family as strictly tumor-suppressive or oncogenic, but rather suggests that it functions in a context-dependent manner. Initially, reports describing miR-26 as a tumor suppressor because of its negative impact on myc-mediated transformation have been supported by studies that demonstrate its down-regulation in diverse cancer entities such as

lymphoma, hepatocellular carcinoma, breast cancer and colorectal carcinoma (39, 40, 41, 42). Typical target genes in this context appear to be regulators of proliferation, such as *Cyclins D2* and *E2* (*Ccnd2* and *Ccne2*) as well as *Enhancer of zeste homolog 2* (*EZH2*), an epigenetic modifier with a widespread role in gene regulation during development and differentiation (43, 44, 45). On the other hand, clear oncogenic properties of the miR-26 family have been described in glioma patients that exhibit elevated miR-26 levels accompanied by repression of well-established tumor-suppressors *Pten* and *Rb1* (46). Likewise, miR-26–mediated *Pten* down-regulation has been reported for T cell acute lymphoblastic leukemia (T ALL) and lung cancer (47, 48). Notably, a direct repression of *Pten* by miR-26a has also been documented in WEHI-231 progenitor B cells, and it has been postulated that this miR-26a/b–mediated reduction of *Pten* can evade central tolerance mechanisms by promoting the survival of autoreactive B cells (49). Thus, it appears that cell intrinsic and extrinsic cues determine whether the tumor-suppressive or oncogenic activity of the miR-26 family dominates in a certain setting, and that the balance between these opposing functions likely differs from tissue to tissue.

Our findings place progenitor B lymphocytes into the latter group, as aberrant overexpression of miR-26 family members supports their expansion, counteracts induced apoptosis, severely blocks the pre-B to immature B cell transition and eventually drives oncogenic transformation, at least to some degree by direct targeting of *Pten*. This in accordance with previous reports identifying PTEN as a prominent tumor suppressor in several tissues, among them T-cell progenitors, in which postnatal *Pten* knockdown produced highly disseminated T-cell acute lymphoblastic leukemia (50, 51). Supporting this, deletions or inactivating mutations of *Pten* have been described in all main types of human cancer. PTEN exerts its tumor suppressive function by dephosphorylating phosphatidylinositol 3,4,5-trisphosphate (PIP3) to PIP2, therefore

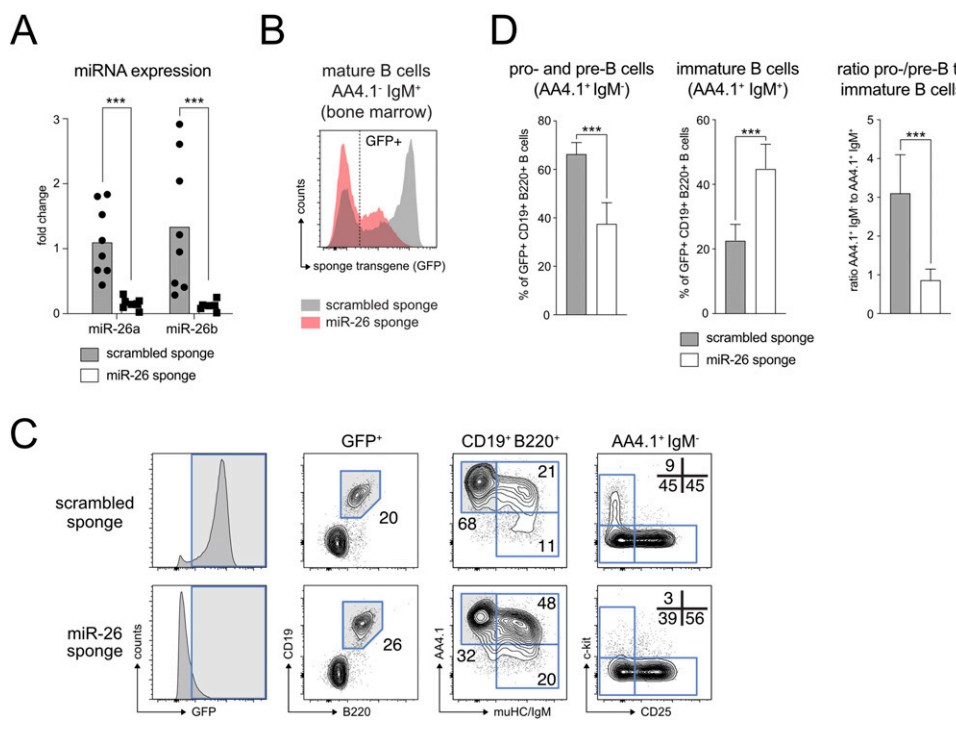

**Figure 6. An in vivo miR-26 loss-of-function mouse model recapitulates the enhanced pre-B cell differentiation phenotype.**
**(A)** GFP+ pro- and pre-B cells as defined by B220$^+$IgM$^-$AA4.1$^+$ of scrambled control or miR-26 sponge mice were FACS sorted and analyzed for expression of miR-26a and b by quantitative PCR. Individual experiments were normalized to the mean expression of scrambled sponge mice. Data are representative of at least eight independent experiments. **(B)** Histogram overlay depicting scrambled (grey) and miR-26 (red) sponge expression in mature bone marrow B cells based on GFP fluorescence. **(C)** Representative gating strategy to evaluate early B cell development in scrambled and miR-26 sponge-expressing mice. Gates that are highlighted in grey define the next population in the hierarchy (from left to right). Pre-gates were set on GFP+ and CD19$^+$B220$^+$ B cells, and pro- and pre-B cells were further defined as AA4.1$^+$IgM$^-$, immature B cells as AA4.1$^+$IgM$^+$, and mature B cells as AA4.1$^-$IgM$^+$. Pro− and pre-B cells were further subdivided into pro−B cells (c-kit$^+$CD25$^-$), pre-B cells (c-kit$^-$CD25$^+$), and an intermediate population (c-kit$^-$CD25$^-$). Numbers represent the percentage of cells within the respective gates. **(D)** The bar graphs summarize the percentages of pro− and pre-B cells and immature B cells among GFP+ B cells, as well as the ratio of those values, in control and miR-26 sponge mice. Bars show mean + SD and represent at least eight independent experiments. ***$P < 0.001$.

counteracting PI3K and its downstream effectors such as AKT. In early B cell development, PI3K signaling is essential to ensure pre-B cell survival, and its precise regulation coordinates the step-wise cellular progression through phases of proliferation and differentiation (10, 32, 52, 53). PI3K activation in this context appears to be mainly initiated from the pre-BCR (53, 54), which signals via the co-receptors BCAP and CD19, possibly with a contribution from the IL-7 receptor (10). This is of particular interest in the light of our findings that aberrant expression of miR-26a arrests pre-B cells at a pre-BCR-positive state, and moreover, that pre-BCR-derived signals remain essential even after transformation. It is tempting to speculate that the strong AKT phosphorylation that we observe is mediated by two mechanisms: One the one hand, by direct repression of *Pten*, and on the other hand by ensuring continuous PI3K pathway activation through its upstream receptor. Notably, however, there appears to be an upper threshold for PI3K pathway activation that is tolerable for B cells, as activating mutations within the regulatory subunit p85 alpha severely compromise B cell development and function (55). Along the same line, complete or partial loss of *Pten* and the resulting hyperactivation of the PI3K pathway in a pre-B cell ALL mouse model has been shown to result in rapid p53-mediated cell death of leukemic cells (56). As such, PTEN may play a different role in pre-B ALL compared with other hematopoietic malignancies, and supporting this, PTEN levels in pre-B ALL appear higher than, for example, in lymphoma (56). Although this may appear incompatible with our own findings, several points indicate that this model is not directly comparable with the miR-26a overexpression phenotype. First, *Pten* loss in pre-B ALL was accompanied by down-regulation of the IL-7R, CD19, and of pre-BCR

components (56), all of which was not observed in our system. Moreover, the reduction in *Pten* expression and the concomitant increase in AKT activation were not counterselected in our hands, but cells expressing miR-26a demonstrated a clear competitive advantage over controls. A possible explanation for this discrepancy may be that miR-26–mediated transcriptional deregulation affects multiple genes, both directly and indirectly. It is therefore difficult to compare cells with reduced expression of *Pten* to the more complex miR-26 gain-of-function phenotype. In fact, even if *Pten* repression is detrimental, it appears conceivable that one or several of those genes affected by miR-26a can compensate for this effect. Alternatively, it is possible that the upper threshold of PI3K activation is simply not exceeded in our system. The unusual oncogenic role of PTEN in the pre-B ALL model was described in the context of Bcr/Abl- and NRas-mediated leukemia initiation, both strong oncogenes on their own. It may very well be that it is the combination of increased PI3K activation together with such potent oncogenic drivers that pushes the cells into a p53-mediated cell death. However, this clearly needs to be addressed in follow-up studies.

Beyond the oncogenic gain-of-function setting, our findings indicate that loss of miR-26 expression also controls early B cell development and expansion, albeit in a reciprocal manner. In particular, cells lacking miR-26 family members tend to proliferate less well and appear poised towards pre-B to immature B cell differentiation. As such, it is tempting to postulate a rheostat-like role for the miR-26 family in progenitor B cells: On one end of the spectrum, an increase in mature miR-26 family members over basal levels signals cell expansion, and on the other end, a reduction

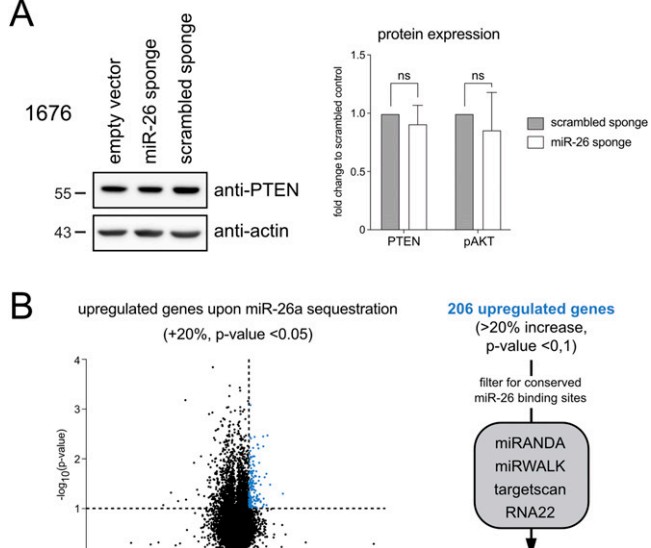

**Figure 7. Transcriptome analysis does not identify clear target genes upon miR-26 sequestration.**
**(A)** Western blot analysis comparing PTEN protein expression in 1676 pre-B cells expressing the miR-26 sponge or a scrambled control. Beta-actin serves as a loading control. The Bar chart shows the statistical analysis for PTEN and phosho-AKT. **(B)** A volcano plot comparing the transcriptional landscape in miR-26 sponge-expressing 1676 pre-B cells and the respective control (two independent experiments). The horizontal dashed line marks the *P*-value cutoff (*P* < 0.1), the vertical line depicts an up-regulation by 20% or more. Genes that are significantly up-regulated in response to miR-26 sequestration are displayed in blue. The flow chart on the right summarizes the strategy for miR-26a and b target identification in analogy to Fig 4A.

facilitates progression of developing B cells from the pre-B compartment to the immature B cell stage. Notably, the molecular programs that need to be initiated at the pro-B cell stage and at the pre-B cells stage are similar, with a proliferative phase that is followed by a cell cycle arrest in which V(D)J recombination takes place. As such, it is possible that the miR-26–mediated regulation described here extends also to pro-B cells. Indeed, pro-B cells in the loss-of-function mouse model were also strongly decreased, indicating that both pro- and pre-B cells lacking miR-26 activity are poised to progress towards the next developmental stage.

How the miR-26 family establishes this rheostat-like function needs to be addressed in more detail. In the gain-of-function approach, we have identified *Pten* as one of the targets likely involved in the phenotype, as its knockdown recapitulated the block in pre-B cell differentiation. In addition, our analysis retrieved candidates such as Skp1-Cul1-F-box (SCF) E3 ubiquitin ligase *Fbxl19* and the pseudokinase *Stradb*, both of which have been recently identified as direct targets of miR-26 in adipocytes (37). Knockdown studies of *Fbxl19* in lung epithelial cells have indicated a negative role in cell proliferation, making it is possible that its loss enforces the pro-proliferative response upon miR-26 overexpression (57). STRADB, on the other hand, regulates LKB1 kinase, which has been shown to synergize with PTEN to exert its tumor-suppressive role in several cancer entities (58, 59, 60). Thus, it will be interesting to

evaluate whether STRADB may also play a role in miR-26–mediated regulation of pre-B cells.

In the loss-of-function approach, we failed to detect clear target genes of the miR-26 family, which was surprising because of the fact that the sponge mouse as well as the in vitro experiments displayed the reciprocal phenotype to the aberrant overexpression. Moreover, we were able to confirm miRNA sponge function by TDMD-mediated loss of mature miRNAs miR-26a and b in vivo, as well as by de-repression of a reporter for endogenous miR-26 family members in vitro. The lack of clear targets in this approach is therefore difficult to explain. Most likely, endogenous miR-26a and b exert their functions not by regulating one specific gene but rather repress several target gene networks or pathways. Thus, it is possible that the overall phenotype is established by subtle changes of several genes, maybe even with a large cell-to-cell variation, which may not be identified by our transcriptome approach. In this context, one also needs to keep in mind that the sponge approach does not reflect a complete loss-of-function, but rather reduces endogenous miRNA levels. As such, residual amounts of miR-26a and b may be sufficient to mask miRNA targets. As an alternative approach to the miR-26 sponge, which we generated to circumvent the tedious targeting of three individual miRNA loci, a recent study has used individual sgRNAs to disrupt the function of all miR-26 family members in ES cells (37). Whereas the authors do not comment on any phenotype beyond the prominent defect in adipocyte regulation, it will be interesting to investigate whether this cleaner approach recapitulates our findings about the regulatory role of the miR-26 family in early B cell development.

# Materials and Methods

### Ethics statement

Experimental procedures with animals were approved by the institutional ethics and animal welfare committees of the University of Veterinary Medicine Vienna and the Medical University of Innsbruck in accordance with good scientific practice guidelines and national legislation (license numbers: BMBWF-68.205/0023-II/3b/2014 and BMBWF-66.011/0021-V/3b/2019).

### Mice

The miR-26 sponge mouse and the scrambled control mouse model were generated by flippase recombinase (FLP)-mediated knock-in of a CAG promoter-driven loxP-STOP-loxP cassette into the safe harbor locus ColA1 (61). In short, transgenic cassettes consisting of a EGFP cDNA and concatemeric repeats of miR-26a/b binding sites (gt**AGCCTATCCTCTATACTTGAA**cc x26; binding site in bold) or scrambled binding sites (gt**TTAGAATTTAAACCTCACCATGA**cc x27; scrambled site in bold) in the 3′-UTR were cloned into CAGGS-loxSTOPlox-ClaI flp-in (reference 61; Addgene plasmid #21548, kindly provided by Rudolf Jaenisch) and then electroporated together with a construct encoding FLPe into KH2 ES cells (reference 62; C57BL/6 x 129/Sv background, kindly provided by J. Zuber, IMP). ES cell clones selected with hygromycin B (Carl-Roth) and screened

for correct transgene insertion by PCR were used for injection in C57BL/6NRj blastocysts. High percentage chimeras were bred with C57BL/6NRj females to confirm germline transmission and then further backcrossed to generate a congenic strain. To delete the loxP-STOP-loxP cassette and thus to induce transgene expression, miR-26 and scrambled sponge mice were crossed with a CMV-Cre strain (C57BL/6N-Tg(CMV-cre)1Cgn) in which the cre gene is under control of a human cytomegalovirus minimal promoter (63). For all analyses, the CMV-Cre transgene was then crossed out and the sponge transgenes were bred to homozygosity.

Mice were kept in a facility for laboratory rodents under specific pathogen-free and controlled environmental conditions (temperature 22°C ± 1°C, relative humidity of 40–60%, 12:12-h light/dark cycle) according to FELASA recommendations (64). Food (regular mouse diet (Ssniff)) and water were provided ad libitum. Animals were maintained in small groups in individually ventilated cages lined with wood shavings and enriched with nesting material. If not stated otherwise, mice were analyzed at an age of 10–12 wk. For all experiments, male and female mice were used in comparable frequencies.

### Pre-B cell lines

The pre-B cell lines 1676, wk3, and TKO were derived by extended culture of total bone marrow of $SLP\text{-}65^{-/-}LAT^{-/-}$, $SLP\text{-}65^{-/-}$, or $Igll1^{-/-}Rag2^{-/-}SLP\text{-}65^{-/-}$ (TKO) mice, respectively, in IL-7–supplemented medium and have been previously described (23, 65). Pre-B cell lines, primary bone marrow–derived pre-B cells, and 293T cells were cultured in IMDM (Sigma-Aldrich) containing 7.5% FBS (Superior; Sigma-Aldrich), 100 U/ml penicillin, 100 U/ml streptomycin (Sigma-Aldrich), and 50 $\mu$M 2-ME. For pre-B cells, this medium was supplemented with the supernatant of IL-7–expressing J558L cells in excess if not stated otherwise. The Syk kinase was inhibited using compound R406 (Selleckchem) at a concentration of 2 $\mu$M. For analysis of DNA damage induced apoptosis, cells were treated with 5 nM Etoposide for 20 h.

### Plasmids, transfections, and viral transductions

The miRNA expression library, the miRNA sponge constructs, and the vector encoding constitutively active Akt (myr-Akt) have previously been described (23, 27, 66). For knockdown of PTEN and CD79b, shRNAs encoding the specific 21-mer guide strands were cloned into the miR-30-adapted LMP vector (sequences of the guide strands are listed in the Supplemental Data 1).

Retroviral supernatants were produced in HEK293T cells. Per sample, 400 ng of DNA for the plasmid of interest, 150 ng of HIT60, and 150 ng pVSVg were diluted in IMDM (Sigma-Aldrich) and mixed with IMDM containing polyethylenimine (PEI, Polysciences) in a 1:3 DNA:PEI ratio. After a 20-min incubation step at room temperature, the DNA:PEI mix was plated with 2 × 10⁵ HEK293T cells in a total volume of 250 $\mu$l. After 30–40 h, viral supernatants were harvested, mixed with polybrene (16 $\mu$g/ml final concentration), and used for spin-infection (400$g$ at 37°C for 60–90 min) of cells.

### Microarray analysis

Total RNA was isolated (RNeasy system; QIAGEN) from 1676 pre-B cells cultured in the presence of IL-7 and selected for expression of

an empty control vector, an miR-26a construct or the miR-26 sponge. The processing of RNA samples, microarray hybridization (Agilent G3 Mouse 8 × 60K array), and data quality control were performed by IMGM (Martinsried). Data were analyzed using the Subio software platform (Subio). In short, data files for two biological replicates were quantile-normalized, log₂-transformed, normalized to the control sample of each experiment, and averaged. Measurements that did not exceed background levels in all six samples or whose Ch1 raw signal did not exceed a mean value of 50 in at least two four of the six samples were discarded. For the identification of down- or up-regulated genes, pairs of sample groups were filtered for annotated genes whose expression was reduced by at least 20% ($P$-value < 0.05) upon miR-26a overexpression or increased by 20% ($P$-value < 0.1) upon sponge-mediated sequestration. For the miRNA target identification, these lists of genes were further filtered against computationally predicted miR-26a and b targets exploiting the miRWalk2.0 atlas (67). Genes were defined as targets if predicted by at least two of the four algorithms miRWalk, miRanda, RNA22, and Targetscan.

### Flow cytometry

For bone marrow cell suspensions, femurs and tibiae were isolated, ground, and filtered through a 70-$\mu$m mesh. Cells were resuspended and washed in FACS buffer (PBS with 1% FCS). Single-cell suspensions were stained in 96-well plates with 30 $\mu$l of the respective antibody cocktails for 20 min at 4°C. All centrifugation steps were performed with 530$g$ for 2 min. For the antibody cocktails, the following antibodies were used: anti-B220-BV510 (RA3-6B2; BioLegend), anti-CD19-BV605 (6D5; BioLegend), anti-CD93-PE/Cy7 (AA4.1; BioLegend), anti-CD93-APC (AA4.1; BioLegend), anti-CD1d-PE (1B1; eBiosciences), anti-CD25-PE (PC61; BioLegend), anti-cKit-APC (2B8; BioLegend), anti-IgD-PerCpCy5.5 (11-26C.2A; BioLegend), anti-IgM-eFluor450 (eB121-15F9; eBioscience), anti-CD79b-FITC (HM79-12; BioLegend), anti-KLC-bio (RMK-12; BioLegend), anti-CD179b-bio (LM34; BD Biosciences), and anti-Streptavidin-FITC (eBiosciences). Apoptotic cells were stained with DAPI or PI (1:10,000) for exclusion of dead cells and AnnexinV-FITC (BioLegend) before flow cytometric analysis. Alternatively, apoptotic cells were quantified by flow cytometry using the Vybrant Apoptosis Assay Kit (Thermo Fisher Scientific). To measure proliferation, cells were labeled with 10 $\mu$M EdU for 1 h and then stained using the Click-IT EdU Flow Cytometry Assay kit (Invitrogen) according to the manufacturer's recommendation. To assess the DNA content, cells were fixed and permeabilized in 70% ethanol on ice and stained with 1.6 $\mu$g/ml propidium iodide and 1 $\mu$g/ml RNase in PBS for 30 min at 37°C. The relative fitness of transduced versus non-transduced cells within the same sample was analyzed based on a competitive growth assay (68). Changes in fluorescent marker expression over time were converted into a growth index with an arbitrary cell doubling time of 1 d.

### Quantitative real-time PCR

For RNA isolation, sorted or suspension cells were pelleted and resuspended in 500 $\mu$l TRIzol reagent (15596026; Thermo Fisher Scientific). Total RNA was isolated according to manufacturer's instructions. RNA was reverse-transcribed using random hexamer primers followed by SYBR green-based quantitative PCR. Oligo sequences are

listed in the Supplemental Data 1. For miRNA quantification cDNA synthesis was performed using specific primers (hsa-miR-26a 000405, has-miR-26b 000407, snoRNA202 001232; Thermo Fisher Scientific), followed by probe-based qPCR (AceQ qPCR Master Mix Q112-02; Vazyme). Final quantification was performed using the ddCT method. Note that only individual data points, but no error bars, are included in the figure panels that display the fold change and that the statistical significance has been calculated based on the ddCT values.

### Western blot

For Western blot analysis, cells were lysed in ice-cold RIPA buffer (50 mM Tris–HCl, pH 7.4, 1% NP-40, 0.25% sodium deoxycholate, 150 mM NaCl, 1 mM EDTA [pH 8], protease inhibitor cocktail [Sigma-Aldrich], 10 mM Na$_3$VO$_4$, and 10 mM NaF), mixed 1:1 with reducing sample buffer (62.5 mM Tris pH 6.8, 2% SDS, 10% Glycerol, 100 mM DTT), boiled for 5 min at 95°C, and loaded onto an SDS–PAGE gel. Western blotting was performed using PVDF membranes. Proteins were detected using anti-PTEN (D4.3), anti-phosphoAkt Ser473 (D9E), anti-Akt (pan; C67E7), anti-p38 (D13E1), anti-actin (13E5; all Cell Signaling Technologies), and anti-phosphotyrosine (4G10; Merck) antibodies. Immunoreactive proteins were visualized with HRP-labeled secondary antibodies and the ECL system (Advansta) on light-sensitive film (Amersham, GE).

## Data Availability

Microarray data have been deposited under GEO accession number GSE186236.

## Supplementary Information

## Acknowledgements

We thank K Rossi, S Gritsch, and I Gaggl for animal care and their technical assistance. This work was supported through the Tiroler Wissenschaftsfond (TWF) and the Austrian Science Fund (FWF; P30196-B26) to S Herzog.

### Author Contributions

K Hutter: formal analysis, validation, investigation, visualization, methodology, and writing—original draft, review, and editing.
SE Lindner: formal analysis, validation, investigation, visualization, and methodology.
C Kurschat: investigation and methodology.
T Rülicke: resources, funding acquisition, and writing—original draft.
A Villunger: resources, funding acquisition, and writing—original draft.
S Herzog: conceptualization, data curation, formal analysis, supervision, funding acquisition, investigation, visualization, methodology, project administration, and writing—review and editing.

### Conflict of Interest Statement

The authors declare that they have no conflict of interest.

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
