## [Reviewer comments · Life Science Alliance]

Life Science Alliance

The miR-26 family regulates early B cell development and transformation

Sebastian Herzog, Katharina Hutter, Silke Lindner, Constanze Kurschat, Thomas Rüllicke, and Andreas Villunger

DOI: <https://doi.org/10.26508/lsa.202101303>

Corresponding author(s): Sebastian Herzog, Innsbruck Medical University

Review Timeline:

Submission Date:	2021-11-18
Editorial Decision:	2021-12-20
Revision Received:	2022-03-09
Editorial Decision:	2022-03-30
Revision Received:	2022-04-07
Accepted:	2022-04-08

Transaction Report:

December 20, 2021

Re: Life Science Alliance manuscript #LSA-2021-01303-T

Dr. Sebastian Herzog
Innsbruck Medical University
Division of Developmental Immunology
Innrain 80
Innsbruck 6020
Austria

Dear Dr. Herzog,

Thank you for submitting your manuscript entitled "The miR-26 family regulates early B cell development and transformation" to Life Science Alliance. The manuscript was assessed by expert reviewers, whose comments are appended to this letter. We invite you to submit a revised manuscript addressing the Reviewer comments.

Thank you for this interesting contribution to Life Science Alliance. We are looking forward to receiving your revised manuscript.

Sincerely,

B. MANUSCRIPT ORGANIZATION AND FORMATTING:

Reviewer #1 (Comments to the Authors (Required)):

Manuscript by Hutter et al presents novel insight into the role of miR-26/PTEN axis in early B cell development. The authors demonstrate that miR-26 directly targets PTEN in developing B lymphocytes using over-expression of this miRNA to show that augmented expression of this miR promotes block in differentiation. Intriguingly, they show that enhanced expression of miR-26 and augmented PTEN expression renders the cells independent of IL7 and suggest that this promotes their proliferation. In a different line of experiments, the authors utilize a sponge system to demonstrate that diminished miR-26 renders cells less fit to compete with WT B progenitors, showing that "transformed" cells may be dependent on persistent miR-26 signaling - perhaps being addicted to the reduced levels of PTEN and therefore higher levels of PI3K. Intriguingly, knockdown of endogenous miR-26 using target-directed miR degradation enhances differentiation of progenitor B cells.

Overall, the manuscript is compelling and appropriate for LSA and will be of interest to a broad audience, including immunologists and cancer biologists. There are a few points that should be addressed:

1. There is a prior publication that hints at the role of miR-26 in the regulation of PI3K signaling/Pten - a Nature Imm report from 2016 by Gonzalez-Martin et al. shows that miR-26a targets PTEN in WEHI-231 B cell line. While the present manuscript goes into much more mechanistic detail and provides much novel insight, the 2016 publication should be appropriately acknowledged.
2. In vivo studies, such as those presented in Fig. 6 should include absolute counts, and not just relative % for B cell subsets to substantiate the statements about the impact of miR-26 on enhancing/accelerating differentiation of pre-B cells.
3. For experiments in Fig. 1 - is there any direct data on proliferation? The authors make a point of augmented proliferation being responsible for the phenotype solely based on lack of impact on survival, perhaps a direct assessment of proliferation would let them make a stronger point.
4. In systems where miR-26 is overexpressed and therefore PI3K signaling is augmented, is there any evidence of Ig deficient B cells persisting in the periphery - as BCR tonic signal is dependent on PI3K, perhaps some of the accelerated differentiation is due to this?
5. Can the lack of evident difference in PTEN (or really other readily recognizable miR-26 targets) in Fig 7 be due to the cells simply not tolerating high PTEN expression and subsequent diminished PI3K signaling?

Reviewer #2 (Comments to the Authors (Required)):

The manuscript of Hutter et al. on "The miR-26 family regulates early B cell development and transformation" studies the function and the role of the small micro-RNA miR-26a in early B cell development. The starting point of this study was a screen for micro RNAs which blocked the transition from pre-B cells to immature B cells induced by IL-7 withdrawal. In this screen the authors show that overexpression of miR-26 blocks B cell maturation and they confirm this result in a more detailed analysis. In a search for pre-B cell expressed genes, which are affected by overexpression of miR-26, the authors identify, among others, PTEN as a candidate which is regulated by this micro-RNA. It is well known that PI3 kinase signaling is promoting pre-B cell expansion and that the switch from pre-B cell to immature B cells is accompanied by reduced PI3 kinase signaling and increased expression of PTEN, a lipid phosphatase that inhibits the PI3 kinase signaling pathway. Thus, the identification of PTEN as a target gene for miR-26 fits to the observed phenotype. However, given the fact that the reduction of PTEN expression is variable, it is not completely clear whether PTEN is the dominant miR-26 target or whether other genes also are affected by overexpression of miR-26 or vice versa. I therefore think it would be important to show whether or not an overexpressing of PTEN to different levels, counteracts the blocking effect of miR-26 in the pre-B to immature B cell transition. Otherwise, this study is interesting and worthwhile to be published in the Life Alliance Journal.

Specific comments:

1. In their study of miR-26 function the authors are using different pre-B cell lines or primary B cell progenitors, but it is not always clear what lines are used. Thus, it would be helpful for the reader if the authors indicate the use of the cells not only in the figure legends, but also in the figure itself. Furthermore, what does it mean when they write in the figure legend, for example under Fig. 1c they use Wk3 pre-B cells or primary B cell progenitors? For which figure, Fig. 1D or 1E do the authors use primary B cell progenitors? Furthermore, it would be important that the authors state why they do some experiments with 1676 pre-B cell line and others with the Wk3 line.
2. The survival and anti-apoptosis effect of miR-26 is shown in Fig 1 E and F. Fig 1E shows the pro-survival effect mediated by miR-26. Both results {plus minus} miR-26 are within the normal apoptosis range of cultivated cell lines, so it would be desirable to repeat this experiment (N=?) and perform a significance calculation. For the calculation of the living cell population in Fig 1F, it should be shown, how the gates were set in the FACS plot.
3. The anti phospho-tyrosine blot shown in Fig. 2C is not very informative without knowing what phosphorylated substrates are analyzed in this blot. The direct connection between miR-26 expression and PTEN is also not clear if one regards the result of the removal of the miR-26 by a specific sponge where the PTEN expression is not really increased. Although the introduced sponge clearly increases the pre-B cell to immature B cell transition, these data suggest that also other genes (or other mi-Rs e.g., miR-19, miR-19~92, miR-150, please comment) are involved in the regulation of this process and it would be helpful to learn more about those, if possible. In the differentiation studies shown in Fig. 6C it would be interesting whether the authors extend their studies not only on the IgM-BCR, but also on the IgD-BCR and the lambda light chain isotype.
4. The authors stated in their discussion: " miR-26- mediated PTEN downregulation has been reported for T cell acute lymphoblastic leukemia (T ALL) and lung cancer". It would be interesting how they explain the fact that miR-26a was found down-regulated in all patients with B-ALL (Cancer Biomark. 2015;15(3):299-310. doi: 10.3233/CBM-150465) when they relate the findings to pre-B cells leukemia.

Minor points:

1. In the introduction the authors use the term "light chain recombination" or rearrangement of light chain gene segments. I think they should here stick to the proper nomenclature. There is no rearrangement of light chains, but of light chain variable genes. Furthermore, the rearrangement process involves V gene segments and not light chain gene segments.
2. Fig 5 A shows that inhibition with miR-26 sponge increases the population of differentiated cells. The number of cells measured in miR-26 sponge/dsRed+ seems to be significantly higher than in the other gates. Although the percentage distribution would be unaffected, it is more reliable to measure comparable cell numbers.
3. Some experiments and data are "not shown". It would be desirable to include them in the supplemental part.
4. K.K. and S.H. wrote Please correct author contributions

Dear Editors, dear Reviewers,

we would like to thank you for the time you invested into critically reading and evaluating our manuscript "The miR-26 family regulates early B cell development and transformation", which we have submitted for publication as an article in *Life Science Alliance*. We appreciate the positive comments about our work and the insightful suggestions that enabled us to improve the quality of our manuscript.

Although not requested by the Reviewers, please note that in the revised manuscript we have provided quantitative PCR data showing that pre-B cells expressing miR-26 fail to induce critical genes such as *Rag1*, *Rag2* and *Aiolos* upon IL-7 withdrawal (suppl. Fig. 1A), which corroborates the block in differentiation as measured by kappa light chain expression.

That said, in the following pages we address each concern raised by the Reviewers point by point.

Reviewer 1

Comment: There is a prior publication that hints at the role of miR-26 in the regulation of PI3K signaling/Pten - a Nature Imm report from 2016 by Gonzalez-Martin et al. shows that miR-26a targets PTEN in WEHI-231 B cell line. While the present manuscript goes into much more mechanistic detail and provides much novel insight, the 2016 publication should be appropriately acknowledged.

Response: We thank this Reviewer for this suggestion. Indeed, while the key finding of this study is the identification of miR-148 as a critical regulator of central tolerance, the experimental approach with miRNA pools that were used to transduce HSCs also retrieved miR-26a and b as tolerance breaking. As this Reviewer pointed out, in the supplementary data the authors show that overexpression of miR-26a results in repression of PTEN in WEHI-231 cells, which is in accordance with our own findings. To acknowledge this study, we have included its reference in the discussion.

Comment: *In vivo* studies, such as those presented in Fig. 6 should include absolute counts, and not just relative % for B cell subsets to substantiate the statements about the impact of miR-26 on enhancing/accelerating differentiation of pre-B cells.

Response: We agree with this Reviewer that absolute cell counts in addition to relative percentages should in general be provided with all in vivo experiments. In this particular case, however, we would like to point out that only a subset of cells within our miR-26 sponge model and the respective control express the transgene (see also Figs. 6B and C). Moreover, the scrambled sponge mouse displays significantly higher portion of GFP⁺ cells

compared to the miR-26 sponge, and the relative portion of GFP+ cells varies from stage to stage (see below). Thus, we think that absolute numbers for the respective GFP+ populations provide no additional information.

Of note, displaying the percentage of GFP+ cells in the different stages clearly indicates that GFP+ cells become significantly enriched at the pre-B to immature B cell transition in miR-26 sponge mice compared to the control, which complements our data presented in Fig. 6B to D. If this Reviewer thinks that such a graph would be beneficial for our manuscript, we are of course happy to add it to the main figure or as a supplement.

Comment: For experiments in Fig. 1 - is there any direct data on proliferation? The authors make a point of augmented proliferation being responsible for the phenotype solely based on lack of impact on survival, perhaps a direct assessment of proliferation would let them make a stronger point.

Response: This is an important point. While the competition assay that monitors enrichment or loss of the transduced population over time is very sensitive, it cannot distinguish between cell survival and proliferation effects. We therefore have directly assessed cell proliferation through different means, including staining of cells with a dye that is diluted with each division, DNA content staining as well as EdU labeling. The first approach turned out to be not feasible under steady state conditions, i.e. in fast cycling cells that do not require any stimulation. DNA content staining and EdU labeling, on the other hand, did not reveal a clear proliferative advantage of cells overexpressing miR-26 when IL-7 was provided in excess, but it may very well be that these techniques are just not sensitive enough. However, since we cannot claim that miR-26 promotes proliferation based on these data, we have rewritten this section and provide these new findings, together with an analysis on apoptosis under steady state conditions, in the novel supplementary Fig. S1.

Comment: In systems where miR-26 is overexpressed and therefore PI3K signaling is augmented, is there any evidence of Ig deficient B cells persisting in the periphery - as BCR tonic signal is dependent on PI3K, perhaps some of the accelerated differentiation is due to this?

Response: It is an interesting hypothesis that enhanced expression of miR-26 and the concomitant increase in PI3K signaling may allow the persistence of BCR-negative mature cells in the periphery. As we have not done this type of overexpression experiment in vivo, but only the opposite loss-of-function approach, we cannot address this point. To our knowledge, however, no such cells have been reported by other studies, in particular by Zeitels et al..

With respect to the accelerated differentiation that we see upon loss of miR-26 expression, i.e. in the reciprocal scenario, one can speculate that reduced PI3K activity at the pre-B cell stage may limit the normal proliferative burst and thus enable a “premature” differentiation. In fact and as discussed in this manuscript, we have shown before that inhibition or dampening of PI3K signaling primes cells for the pre-B to immature B cell transition.

Comment: Can the lack of evident difference in PTEN (or really other readily recognizable miR-26 targets) in Fig 7 be due to the cells simply not tolerating high PTEN expression and subsequent diminished PI3K signaling?

Response: Given that we see that miR-26 sponge-expressing cells are outcompeted over time in vitro, it is possible that one may end up with cells expressing the transgene only to an extent that does not provoke any “meaningful” gene repression. However, it is surprising that levels of miR-26a and b are nevertheless significantly reduced in the sponge mice in vivo, and that we see the reciprocal phenotype to the miR-26 overexpression. Clearly, more work is needed to decipher how loss of miR-26 affects the transcriptional landscape at the pre-B cell stage.

Reviewer 2

Comment: In their study of miR-26 function the authors are using different pre-B cell lines or primary B cell progenitors, but it is not always clear what lines are used. Thus, it would be helpful for the reader if the authors indicate the use of the cells not only in the figure legends, but also in the figure itself. Furthermore, what does it mean when they write in the figure legend, for example under Fig. 1c they use Wk3 pre-B cells or primary B cell progenitors? For which figure, Fig. 1D or 1E do the authors use primary B cell progenitors? Furthermore, it would be important that the authors state why they do some experiments with 1676 pre-B cell line and others with the Wk3 line.

Response: We apologize for the unclear labeling of the figures. As suggested, we now provide labels for all figures, and state more clearly in the figure legends which type of cells were used in the respective panel. In case of Figure 1C to E, the primary data (upper panels in the respective figure) were generated in 1676 cells, but the same type of experiment was repeated with primary, bone marrow-derived pro-/pre-B cells and the statistical analysis of these experiments is provided in the bar graphs below. Of note, Fig. 1C was accidentally mislabeled as wk3 and has been relabeled to 1676. In terms of the wk3 cell line, we have added the corresponding data to Figures 1C and D in order to strengthen our point with an additional independent pre-B cell line.

Comment: The survival and anti-apoptosis effect of miR-26 is shown in Fig 1 E and F. Fig 1E shows the pro-survival effect mediated by miR-26. Both results {plus minus} miR-26 are within the normal apoptosis range of cultivated cell lines, so it would be desirable to repeat this experiment (N=?) and perform a significance calculation. For the calculation of the living cell population in Fig 1F, it should be shown, how the gates were set in the FACS plot.

Response: We agree with this reviewer that total effect on apoptosis was weak in the experiments shown in Fig. 1E. However, the relative protective effect mediated by miR-26 was nevertheless clearly significant (see bar graph below the primary data in the original figure; n=8). Still, we decided to repeat these experiments with a freshly thawed batch of cells and now show a higher basal apoptosis rate upon IL-7 withdrawal, but the same or an even stronger pro-survival effect by miR-26 (Fig. 1E). As requested by this Reviewer, we have furthermore added the gates to the FSC-SSC plots in Fig. 1F.

Comment: The anti phospho-tyrosine blot shown in Fig. 2C is not very informative without knowing what phosphorylated substrates are analyzed in this blot. The direct connection between miR-26 expression and PTEN is also not clear if one regards the result of the removal of the miR-26 by a specific sponge where the PTEN expression is not really increased. Although the introduced sponge clearly increases the pre-B cell to immature B cell transition, these data suggest that also other genes (or other mi-Rs e.g., miR-19, miR-19~92, miR-150, please comment) are involved in the regulation of this process and it would be helpful to learn more about those, if possible. In the differentiation studies shown in Fig. 6C it would be interesting whether the authors extend their studies not only on the IgM-BCR, but also on the IgD-BCR and the lambda light chain isotype.

Response: We agree that it would have been useful to identify the differentially phosphorylated proteins in cells undergoing pre-B to immature B cell differentiation. However, we would like to emphasize that the anti-phosphotyrosine antibody (clone 4G10) used here is well known for its performance, whereas specific anti-phosphotyrosine antibodies often generate only weak signals, in particular under steady state conditions. Thus, the most suitable experimental strategy in this case would have been a mass spectrometric approach, but we think that this is a separate project and beyond the scope of this manuscript. The purpose of this blot was simply to illustrate that the "pattern" of intracellular signaling associated with IL-7 withdrawal, which is likely dominated by signals from the IL-7R and the pre-BCR, is counteracted by miR-26 expression. Clearly, Fig. 2C is not a key figure for the overall storyline, and we are happy to move it to the supplements or to remove it completely if this is requested.

Regarding PTEN and in contrast to the gain-of-function situation, we agree that the data presented do not necessarily support this critical signaling regulator as a putative target in the loss-of-function setting. As we discuss, we hypothesize that the loss-of-function phenotype is established by subtle changes of several genes, possibly including PTEN, but that this may not be identified by our "global" approaches due to large cell-to-cell variation or overall subtle but still functional effects. Alternatively, we speculate that miR-26 may directly co-regulate additional genes that are counteracted by PI3K signaling, and that may be implicated in the pre-B to immature B cell transition. This would explain why PTEN

knockdown recapitulates the miR-26a overexpression effects while at the same time the loss of miR-26 function does establish the reciprocal phenotype apparently independent of PTEN. However, we have not identified such genes in the gain- or in the loss-of-function microarray analysis.

Regarding other coding and non-coding genes involved in early B cell development, we certainly agree that miRNAs of the miR-17-92 cluster are implicated in pro- and possibly also pre-B cell survival and function (Ventura et al., Cell 2008; Lai et al., Nat. Comm. 2016). In fact, our own work using a miRNA sponge library (based on the same principles as shown for miR-26 in this study) has revealed a putative role for the miR-15 family, but also for the miR-17 (miR-17 and miR-20), miR-19 (miR-19a and miR-19b), and miR-25 (miR-25 and miR-92) families in the pre-B to immature B cell transition (Lindner et al., EMBO Rep. 2017). However, we have no evidence that those miRNAs are directly or indirectly regulated by miR-26, and therefore think that this should be addressed in an independent study.

With respect to IgD, we can only state that the cells shown in Fig. 6C were also stained with an antibody against IgD, and that the mature population as defined by AA4.1⁺muHC⁺ was also positive for IgD, as expected. Moreover, we did not see any difference in the overall IgD staining pattern in the scrambled compared to the miR-26 sponge. The only stages where we see a clear phenotype is at the pre-B to immature B cells transition, and both of these stages do not express the deltaHC.

We have to admit that we have not stained for expression of the lambda light chain isotype as an “alternative” cell fate in vivo. In line with the literature, however, previous differentiation assays in the pre-B cell lines 1676 and wk3 as well as in bone marrow-derived primary progenitor B cells have shown that lambda LC expression only plays a minor role.

Comment: The authors stated in their discussion: " miR-26-mediated PTEN downregulation has been reported for T cell acute lymphoblastic leukemia (T ALL) and lung cancer". It would be interesting how they explain the fact that miR-26a was found down-regulated in all patients with B-ALL (Cancer Biomark. 2015;15(3):299-310. doi: 10.3233/CBM-150465) when they relate the findings to pre-B cells leukemia.

*Response: The study this reviewer mentions clearly provides an interesting observation. However, we would like to point out that the aim of this work, according to the authors, was to “to evaluate the presence of a general **circulating miRNA expression profile in plasma samples** from [...] B-ALL patients and controls to define differentially expressed miRNA with potential diagnostic use“. Hence, while our work focuses on the cellular function of miR-26a and b, the interesting work of Luna-Aguirre and colleagues quantified miRNAs in plasma to define a pattern for use as a biomarker. In this context, it appears difficult to evaluate whether the miRNA levels measured in plasma samples correlate well with intracellular miRNAs, as other studies that quantified miRNAs isolated from the bone marrow of leukemia patients have not recapitulated the finding that miR-26a is downregulated in B-ALL. In addition, it is surprising that miR-26a was downregulated in B-ALL across all patients, but that miR-26b was among the most stable miRNA genes and was actually used as an internal reference for miRNA quantifications in plasma.*

That said, while our work suggests that oncogenic transformation should be associated with high levels of miR-26a and/or b, we cannot exclude a reciprocal expression pattern in a complex tumor setting. However, we hypothesize that aberrant expression of miR-26 also confers an oncogenic function in a human B cells.

Minor points:

Comment: In the introduction the authors use the term "light chain recombination" or rearrangement of light chain gene segments. I think they should here stick to the proper nomenclature. There is no rearrangement of light chains, but of light chain variable genes. Furthermore, the rearrangement process involves V gene segments and not light chain gene segments.

Response: We apologize for these imprecise terms and have corrected the respective sections throughout the manuscript.

Comment: Fig 5 A shows that inhibition with miR-26 sponge increases the population of differentiated cells. The number of cells measured in miR-26 sponge/dsRed+ seems to be significantly higher than in the other gates. Although the percentage distribution would be unaffected, it is more reliable to measure comparable cell numbers.

Response: We agree that it would be optimal to compare similar cell numbers in the dsRed- and the dsRed+ populations of one sample. In this particular case (Fig. 5A), the contour plot appears misleading, as the dsRed+ percentage distribution was 16 % for the scrambled sponge and 8 % for the miR-26 sponge, respectively. Thus, in the latter sample the dsRed- population was actually much higher than the dsRed+ population, in contrast to the visual impression. In this context, we would like to emphasize that these percentages depend on the transduction efficiency and thus on the virus titer, which often cannot be precisely adjusted to give rise to a 50:50 ratio. In this particular experiment (n=15 for the 1676 cells), we had a broad range of dsRed+ percentages in the replicates, but this did not alter the biological effect. We consider this as strong evidence that the percentage distribution is not affected by differences in transduction efficiencies, as this Reviewer itself points out.

Comment: Some experiments and data are "not shown". It would be desirable to include them in the supplemental part.

Response: We agree with this Reviewer that most of the data should be made available at least in form of supplementary figures. The novel data panels are now part of supplementary Figure S1 (no clear anti-apoptotic effect of miR-26 overexpression under steady-state conditions), supplementary Figure S2 (sponge-mediated derepression of a reporter sensing the activity of endogenous miR-26a and b), supplementary Figure S3 (normal B cell composition in the splenic compartment of scrambled and miR-26-sponge mice) and supplementary Figure S4 (normal expression of Pten mRNA and protein in 1676 cells expressing the sponges as well as in cells isolated from the sponge mice).

We do not show, however, data where we found no transforming activity, such as in of miR-26a in bone marrow-derived primary B cell precursors expressing miR-26a as well as upon PTEN knockout or myristoylated Akt expression in 1676 cells. These experiments would resemble Figure 2A, except that all cells die out over time. We think that such a data panel would be of limited interest for the readers, and hope that this Reviewer agrees on this.

Comment: K.K. and S.H. wrote Please correct author contributions

Response: We apologize for this mistake and have corrected the author contributions in the revised manuscript.

March 30, 2022

RE: Life Science Alliance Manuscript #LSA-2021-01303-TR

Dr. Sebastian Herzog
Innsbruck Medical University
Division of Developmental Immunology
Innrain 80
Innsbruck 6020
Austria

Dear Dr. Herzog,

Thank you for submitting your revised manuscript entitled "The miR-26 family regulates early B cell development and transformation". We would be happy to publish your paper in Life Science Alliance pending final revisions necessary to meet our formatting guidelines.

- please address Reviewer 1's final comments
- please add the Twitter handle of your host institute/organization as well as your own or/and one of the authors in our system
- the figure legend for Figure 2 mentions a panel D that does not exist
- Please indicate molecular weight next to each protein blot
- all figure legends should only appear in the main manuscript file

A. FINAL FILES:

B. MANUSCRIPT ORGANIZATION AND FORMATTING:

**Submission of a paper that does not conform to Life Science Alliance guidelines will delay the acceptance of your

manuscript.**

The license to publish form must be signed before your manuscript can be sent to production. A link to the electronic license to publish form will be sent to the corresponding author only. Please take a moment to check your funder requirements.

Sincerely,

Reviewer #1 (Comments to the Authors (Required)):

The authors improved the manuscript and did a good job of addressing most of the concerns I had. The new data describing the failure of miR26 high cells to upregulate RAG, Ikaros and Aiolos transcripts is quite striking and adds new dimensionality to the story.

The text could use some minor editing for clarity. For example the new sentences added that references work by Gonzalez-Martin et al appears to be missing one or two words. Similarly, the sentences describing the evaluation of RAG/Ikaros/etc could benefit from a bit of editing for clarity. The precise level of miR overexpression in the experiments in S1 should also be shown.

Reviewer #2 (Comments to the Authors (Required)):

The manuscript of Hutter et al. on "The miR-26 family regulates early B cell development and transformation" studies the function and the role of the small micro-RNA miR-26a in early B cell development. In their new version the authors have now corrected the mistakes and addressed the comments of the reviewer. I thus recommend the publication of this MS in LSA

April 8, 2022

RE: Life Science Alliance Manuscript #LSA-2021-01303-TRR

Dr. Sebastian Herzog
Innsbruck Medical University
Division of Developmental Immunology
Innrain 80
Innsbruck 6020
Austria

Dear Dr. Herzog,

Thank you for submitting your Research Article entitled "The miR-26 family regulates early B cell development and transformation". It is a pleasure to let you know that your manuscript is now accepted for publication in Life Science Alliance. Congratulations on this interesting work.

DISTRIBUTION OF MATERIALS:

Again, congratulations on a very nice paper. I hope you found the review process to be constructive and are pleased with how the manuscript was handled editorially. We look forward to future exciting submissions from your lab.

Sincerely,
